# Locating Collection and Delivery Points Using the *p*-Median Location Problem

Snežana Tadić [1], Mladen Krstić [1], Željko Stević [2,*] and Miloš Veljović [1]

1 Faculty of Transport and Traffic Engineering, University of Belgrade, 11000 Beograd, Serbia
2 Faculty of Transport and Traffic Engineering Doboj, University of East Sarajevo, 71123 Lukavica, Bosnia and Herzegovina
* Correspondence: zeljkostevic88@yahoo.com or zeljko.stevic@sf.ues.rs.ba

**Abstract:** *Background*: Possible solutions to overcome the many challenges of home delivery are collection and delivery points (CDPs). In addition to commercial facilities, the role of CDPs can also be played by users' households, providing a crowd storage service. Key decisions regarding CDPs relate to their location, as well as the allocation of users to selected locations, so that the distance of users from CDPs is minimal. *Methods*: In this paper, the described problem is defined as a *p*-median problem and solved for the area of the city of Belgrade, using the heuristic "greedy" and the simulated annealing algorithm. *Results*: Fifty locations of CDPs were selected and the users allocated to them were distributed in over 950 zones. The individual distances between users and the nearest CDPs and the sum of these distances, multiplied by the number of requests, were obtained. An example of modification of the number of CDPs is presented as a way of obtaining solutions that correspond to different preferences of operators and/or users in terms of their distances from the CDPs. *Conclusions*: User households can be used as CDPs to achieve various benefits. Locating CDPs, i.e., selecting households, can be solved as a *p*-median problem, using a combination of heuristic and metaheuristic algorithms. In addition, by modifying the number of medians, the total and average distances between users and CDPs can be better managed. The main contributions of the paper are the establishment of users' households as potential locations of CDPs, the establishment of a framework for analysis of impact of the number of CDPs on the sum and average distances from the customers, as well as the creation of a basis for upgrading and modifying the model for implementation in the business practice.

**Keywords:** home delivery; collection and delivery points (CDPs); households; locating; *p*-median; "greedy" heuristics; simulated annealing; crowd logistics





## 1. Introduction

Home delivery is increasingly important in the context of the development of new business models and e-commerce. The importance of this service is particularly pronounced in the conditions of the COVID-19 pandemic, during which the traditional way of supplying users is occasionally disabled. These circumstances further accelerate the development of online shopping and delivery to customers.

This segment of the supply chain is often referred to as last-mile delivery. The last-mile distribution represents a challenging, complex, and expensive process, taking into account the territorial dispersion of users, narrow time windows, small volume of deliveries, low degree of consolidation of flows, the problem of absence of users and failed deliveries, etc. To overcome these problems and shortcomings, different home delivery models are being developed from the aspects of [1] the need for order, frequency, ordering and payment system, starting and end points, executors, reception method, security, delivery area, speed and time of performing, return flows, etc., but also different technological solutions for deliveries.

Numerous technological solutions and concepts contribute to the more efficient execution of logistics and supply chains, including their last segment, i.e., blockchain [2], digital twins [3], robots [4], drones [5], etc., but are also increasingly important in the context of achieving sustainability goals [6].

A solution that has attracted increasing attention from researchers and all stakeholders (companies, administration, users) in recent decades is the collection and delivery point (CDP). It represents one of the most frequently applied solutions for delivering goods to users. It contributes to reducing the distance traveled, delivery time, number of vehicles engaged, fuel consumption, and external costs [7]. Purpose-built or existing, most often commercial facilities (trade, catering, etc.) are used as CDPs. On the other hand, the role of CDPs can be played by the users themselves, i.e., their households [8,9]. This avoids the construction of dedicated buildings and space occupation and potentially achieves economic benefits for all stakeholders (lower investment costs, earnings of households that provide storage services, lower cost of delivery compared to delivery to classic CDPs, etc.). Given that non-professional storage of other users' shipments is carried out in the household space, this practice has the characteristics of a crowd storage service. Together with other crowd logistics services (primarily crowd delivery), this service has increasing importance and potential for future development.

Locating the CDPs represents one of the key strategic decisions related to the logistics system for the delivery of goods to the user [7,10], which as such affects the planning, organization, and implementation of numerous other tasks at the strategic, tactical, and operational level. When user households have the role of CDPs, the location problem is reduced to the selection of a predefined number of households that will be CDPs from the set of all households that have the conditions to provide this service in the observed area. It can be solved as a *p*-median problem.

In this paper, a hybrid model for determining *p*-median is defined, which is based on heuristic ("greedy" algorithm for determining *p*-median) and metaheuristic (simulated annealing) algorithms, where the solution of the first algorithm was used as the initial solution in the application of the second algorithm. The goal is to locate CDPs in users' households, allocate other users to selected locations, and minimize the sum of distances between users and CDPs. The applicability of the model is demonstrated on the example of locating CDPs in the city of Belgrade, where the potential locations are the households of the users, which provide the crowd storage service. The main contributions of the paper are defining the concept of using user households as potential locations of CDPs, and locating objects using the defined model and sensitivity analysis, which pointed out the influence of modifications in the number of selected locations according to the different levels of services offered by the operator or requested by the user. This allows better management of the total and average distances between users and CDPs. To the best of the authors' knowledge, the described approach has not been applied in previous research. Thus, the basis for upgrading, i.e., modifying the model and implementing solutions in business practice, in the described or similar problem, was established.

The second section reviews the research on e-commerce, home delivery, crowd storage, collection and delivery points, and the *p*-median location problem, and identifies research gaps in these fields. The third section provides the methodology for problem solving, i.e., it describes the CDP location model. In the fourth section, a case study of the model application for the city of Belgrade is presented. The fifth section is devoted to the discussion of the obtained results. Concluding remarks and future research directions are provided in the final section.

## 2. Background

The importance of home delivery research with the application of CDPs and the concept of crowd logistics to meet the demands generated by e-commerce are ascertained through a review of the relevant literature. Although a lot of research has dealt with this topic, some aspects have not yet been sufficiently explored and are identified below.

### 2.1. E-Commerce and Home Delivery

The advent of the Internet has greatly changed the world, and in recent decades, the development of online shopping and home delivery has been especially significant for the fields of trade and logistics. Online ordering, electronic commerce, and business models that are developing in this area are changing the structure of supply chains and the functioning of logistics systems [10–13]. Instead of the traditional way of supply in retail stores, an increasing number of users electronically order goods that are delivered to their home addresses. These changes were further accelerated by the COVID-19 pandemic [14–16]. In the conditions of preventing people flows and the necessity of performing goods flows, home delivery has a significant role both for the supply of users (especially the most vulnerable) and for the survival of companies that use or provide this service. Moreover, these changes are likely to have a longer-term impact, even after the problems that caused them are overcome, as users get used to new supply models. In addition, research predicts further growth in the volume of e-commerce, and thus home delivery. Thus, by 2026, the share of e-commerce in retail may reach 24% [17]. Therefore, it is necessary to pay more attention to the research of theoretical models for solving various problems and the implementation of practical solutions in the field of e-commerce and home delivery.

Home delivery has been analyzed in the context of the development of e-commerce in numerous papers [10,18–20]. The paper [21] dealt with the main challenges and possibilities of implementing attended delivery. In paper [22], authors analyzed the impact of home delivery on urban freight transport. The paper [23] identified carbon emissions associated with home delivery as one of the main indicators of adverse environmental impact. In paper [24], the authors assessed the impact of user household characteristics and their location on demand for home delivery. In paper [1], the authors carried out a comprehensive structuring of home deliveries and a detailed analysis of the advantages, disadvantages, and applicability of certain delivery models in different circumstances. The paper [25] proposed a decision support framework that aims to improve delivery success rates while reducing costs. In recent years, electronic ordering and delivery of food products has attracted special attention [26,27]. To the best of the authors' knowledge, the concept of the planned use of users' households as CDPs, i.e., systematic application of crowd storage service in home delivery, has not been researched or applied in practice.

### 2.2. Crowd Storage

In recent decades, crowd logistics services have gained more and more importance [28–31]. Crowd logistics is based on the principle of connecting people who require logistics resources and people who offer their resources, that is, the resources of their household [32]. Most often, they use their own means of transport to perform the crowd delivery service, and storage space for crowd storage services. Requests for services and their acceptance are most often performed through mobile applications or social networks [33]. Applying the concept of crowd logistics can bring numerous environmental (reduction in traffic congestion and harmful emissions), social (interactions, community support), and economic benefits (greater variety of goods, faster delivery, more flexibility, lower cost, fewer failed deliveries, etc.) [34], both in urban and rural areas [35]. Logistics capacities and resources of households and individuals are not sufficiently used, nor are their potentials sufficiently explored in the literature [8,9,32]. An exception is research on the application of the crowd delivery service in home delivery [36–39].

In the context of using households as CDPs, the service of crowd storage or crowd-sourced receiving is particularly important, but there are still no studies that deal with this service extensively. Households can receive and temporarily store the goods until the customer takes them over. For this purpose, basements, spare rooms, garages, yards, or any other unused household space that provides the possibility for adequate storage of goods can be used [40]. Until now, this concept has been applied most often after unsuccessful deliveries, when the goods are diverted and temporarily stored by the customer's neighbor (neighborhood delivery). For many users, the absence of their neighbors is one of the

reasons for not using the delivery services [41]. There is simply no one to receive the goods in case of their absence. Delivering goods to neighbors can increase delivery efficiency by up to 15% [42]. This concept is particularly attractive for the elderly, both from a social (free time, involvement in social interactions) and from an economic perspective (possibility of earning) [43]. Companies are increasingly requiring customers to specify an address when ordering goods to which delivery should be redirected when they are not at home. According to research, 80% of online customers would allow a neighbor to receive goods on their behalf [44], so companies are increasingly offering customers this option. Although this represents a certain step towards the inclusion of households in distribution activities in the last mile, we cannot yet speak of a systematically planned and organized use of households as CDPs. Therefore, more attention should be paid to this issue, first in research, and then in practice.

### 2.3. Collection and Delivery Points and Their Positioning

CDPs represent a network of locations where suppliers consolidate and deliver ordered goods, and customers pay, pick up, or return them [45,46]. They are also referred to as collection points, cluster points, pick-up points, pick-own-parcel points, reception points, etc. [1]. They can be used as targeted or alternative delivery destinations [47]. They contribute to cost reduction compared to home delivery and eliminate the risk of failed deliveries [10,48,49]. Their number is growing, especially in Europe and the USA, and growth was further accelerated by the COVID-19 pandemic [50]. There are two types of CDPs [1,8,22,47,51,52]: attended and unattended. In the first case, persons employed in the CDP receive and hand over the goods to the users, and in the second case, the users pick up the goods themselves, usually from lockers or containers, using the order reference code. Although the interest of researchers and companies in CDPs has increased in recent decades, the possibility of using users' households as CDPs has been considered in a small number of papers [8,9].

The location of CDPs, as well as other logistics facilities, determines their effectiveness [53–55] and represents one of the key strategic decisions regarding the home delivery logistics system, which affects the planning, organization, and implementation of numerous other tasks on strategic, tactical, and operational levels. A number of factors influence the location of a CDP [56]: availability, accessibility, security, environmental impact/land occupation, costs, methods of use, regulations, etc. However, often, not all factors are taken into account when locating CDPs [50]. When locations are chosen ad hoc, without a plan and a valid theoretical model, they are usually expensive, inefficient, and sub-optimal [57,58]. A particularly significant aspect in this context is the availability of CDPs for users, and their tendency to use them [48].

In paper [58], the problem of locating CDPs solved as a set covering location problem. In paper [59], CDPs are located by identifying crowding patterns in different locations at different times of the day based on public transport data. In paper [60], the problem of locating CDPs and routing is solved using classical genetic and memetic algorithms. Defining the optimal number, location, and size of CDPs in order to maximize the profit generated by their operation can be expressed as integer linear programming problem, that is, the uncapacitated facility location problem [61]. Reviewing the literature, no research was found in which the selection of the location of CDPs was solved as a *p*-median problem.

### 2.4. Approaches to Solving the p-Median Location Problem

In many papers, heuristic algorithms were used to solve the *p*-median problem [62–64]. The paper [65] compared the performance of different heuristic algorithms applied to solve the *p*-median problem. To solve this location problem, metaheuristic algorithms are also applied, individually or in combination with other algorithms. For this purpose, genetic algorithm [66–68] and tabu search [69,70] were used. The paper [71] defined a modified genetic algorithm with a greedy heuristic for continuous and discrete *p*-median problems. To the best of the authors' knowledge, the simulated annealing algorithm was first applied

to solve the *p*-median problem in the paper [72]. In paper [73], the authors solved this location problem by defining a hybrid model of simulated annealing and tabu search. In paper [74], the authors solved the *p*-median problem using different metaheuristic algorithms (genetic algorithm, iterated local search, particle swarm optimization, simulated annealing, and variable neighborhood search) and compared their results. The paper [75] proposed an algorithm based on high-performance computing, random sampling, and spatial voting for solving large-scale *p*-median problems. Searching the existing literature, no research was found that combines the heuristic algorithm presented in this paper with the simulated annealing algorithm for solving the *p*-median problem.

The *p*-median location problem has found wide application in the field of logistics for locating logistics nodes [76–78], distribution centers [79], etc. However, the applications in the field of home delivery and CDPs location selecting have not been recorded so far.

## 3. Methodology

The service area is divided into a certain number of zones containing users to whom the goods are delivered. It is necessary to locate CDPs and allocate users to each of the selected locations, so that the sum of the distances between users and CDPs is minimal. The given task is defined as a *p*-median location problem. This location problem represents a significant and well-studied non-deterministic polynomial time hardness (NP-hard) problem. It was first formulated in paper [80]. It involves locating a pre-known number of facilities on the network, so as to minimize the distance, travel time, or transport costs from the facility to the user or vice versa. *p*-median and its extensions are useful for modeling in many situations, such as locating industrial plants, warehouses, public institutions, etc. [81], as well as distribution systems [82].

First, it is necessary to determine *p*, the number of CDPs to be located. The key questions and distinctions when making a decision on the number of households that have the role of CDPs, and thus the criteria for differentiating models for solving the problem, are as follows:

- Number and spatial dispersion of households that can and want to play the role of CDPs (all or only some households in the service area have the conditions to be CDPs);
- Household storage capacity (unlimited or limited);
- Priority in decision making (more importance is given to operator or user preferences).

The model is defined under the following assumptions. Potential locations for CDPs are households that are representative of the service zones and that represent the median of users in each of the zones. The representative can accept all incoming shipments. In other words, it is assumed that there is no limit to the storage capacity of CDPs. In addition to storage space, households have adequate access to infrastructure (parking, elevator, etc.) and information and communication systems (which enable communication with suppliers and users), and voluntarily accept the role of CDPs. At least one person is present in households between 8 am and 8 pm who can receive or hand over goods to the user.

The model is defined as follows:

Minimize

$$F = \sum_{i}^{n} . \sum_{j}^{n} f_i d_{ij} x_{ij} \tag{1}$$

subject to:

$$\sum_{j=1}^{n} x_{ij} = 1, \ i = 1, 2, \ldots, n \tag{2}$$

$$\sum_{j=1}^{n} x_{jj} = p \tag{3}$$

$$x_{jj} \geq x_{ij}, \ i, j = 1, 2, \ldots, n; i \neq j \tag{4}$$

$$x_{ij} \in \{0,1\}, \ i, j = 1, 2, \ldots, n \tag{5}$$

where

$n$—number of service zones/potential locations of CDPs;

$$x_{ij} = \begin{cases} 1, & \text{if delivery for users from zone } i \text{ is made in zone } j \\ 0, & \text{otherwise;} \end{cases}$$

$d_{ij}$—distance from the representative of zone $i$ to the representative of zone $j$;
$f_i$—number of requests for delivery in the zone $i$ in the observed period;
$p$—number of CDPs to be located in the service area.

The objective function (1) is the sum of the distances of the representatives of zone from the CDPs and the number of user requests in these zones. Constraint (2) ensures that each zone is served by only one CDP. Constraint (3) indicates that a total of $p$ objects should be located in the entire service area. Constraint (4) indicates that users from the zone where the CDP is located are served in that CDP. Constraint (5) reflects the binary nature of the variables $x_{ij}$.

Problems of smaller dimensions can be efficiently solved as a previously set linear programming problem using simple solvers [8]. However, it is significantly more difficult to solve NP-hard problems of larger dimensions using the same tools due to the impossibility of finding an optimal solution or a long computer operating time [83]. Research in the past two decades has proven that the most effective methods for solving such problems are methods that combine the concepts of various algorithmic approaches [84,85]. To solve the defined problem, a hybrid model based on heuristic ("greedy") and metaheuristic (simulated annealing) algorithms is defined in this paper. The "greedy" heuristic algorithm, proposed in paper [86], is used to determine the $p$-median. This simple algorithm does not have to give an optimal solution, but it is most often satisfactory, i.e., close to optimal. Therefore, it is desirable to apply other methods for solving the same problem, in order to find a better solution, defining the previously obtained solution as initial. Given that it is a model for large-scale problems, metaheuristic algorithms are a suitable tool. In this paper, the simulated annealing technique will be used to "improve" the obtained solution. In papers [87,88], the authors were the first to independently propose a simulated annealing method for solving combinatorial optimization problems, using an analogy with the physical processes of material annealing [82]. Annealing means reducing the temperature of the material, which causes small perturbations, i.e., changes in the position of the material particles, until the state corresponding to the lowest energy. At the beginning, the material is in a state of melting, in order to gradually lower the temperature until the lowest energy state is reached.

In each iteration, the energy change $\Delta E$ is achieved. When $\Delta E < 0$, the energy is lower, and therefore, the configuration of particles with this energy value is set for the new initial configuration.

When $\Delta E > 0$, the energy is increased, but the particle configuration corresponding to this energy value should not be automatically rejected. During the gradual lowering, the so-called thermal equilibrium is reached at some temperatures. This implies that even after a large number of perturbations, it is not possible to significantly reduce the energy.

Higher temperatures correspond to a higher probability of a sudden increase in the energy state. The probability $P$ that the system at temperature $T$ will increase its energy by $\Delta E$ is equal to [72]:

$$P = e^{-\frac{\Delta E}{k_b T}} \tag{6}$$

where $k_b$ is Boltzmann's constant. Although it is $1.380649 \times 10^{-23}$, in many studies [82,89,90], $k_b$ is usually set to 1 in order to simplify the mathematical expression and facilitate the implementation of the algorithm. In that case, the expression has a form which is used in this paper as well:

$$P = e^{-\frac{\Delta E}{T}} \tag{7}$$

In order to decide whether the configuration of particles with a higher energy value is accepted as the new initial one, the value $P$ is compared with a random number $R$ from

the interval [0,1], generated using a uniform distribution. If $P > R$, the new configuration is accepted as the new initial configuration; otherwise, it is rejected. The temperature $T$ is lowered when thermal equilibrium is reached, so the described procedure continues at the new temperature.

The analogy of solving a defined location problem with the described procedure and sizes is as follows. The criterion function, i.e., the sum of the user's distance from the CDPs corresponds to the energy value, which should be minimized. An acceptable solution, i.e., the selected $p$ locations for CDPs corresponds to one configuration of particles, and the replacement of individual locations that play the role of CDP in one iteration with other locations corresponds to particle perturbation. Temperature has the role of a control parameter. Decreasing the parameter $T$ and swapping locations are performed until the stopping criterion is met, which may be a predetermined number of iterations, computer time, solution quality, etc.

The defined hybrid model is shown in Figure 1, and its notation has the following meaning:

$b$—current number of locations in the set of medians (CDPs);

$p$—number of medians (CDPs) that need to be located;

$m$—current solution (vector of currently selected locations);

$m'$—a still unconsidered solution in the neighborhood of the current solution (a solution that differs in only one location compared to the current solution);

$m^*$—the best solution;

$M$—set of considered solutions;

$F(m)$—the value of the criterion function (the sum of the distances between users and CDPs, multiplied by the number of requests) for solution m;

$F^*$—value of the criterion function for the best solution;

$T$—current number of executed iterations;

$k$—the number used to define after which number of iterations it is checked whether the criterion function changes significantly;

$x$—number that defines which change in the criterion function is considered significant;

$c$—constant by which the parameter $T$ is multiplied upon reaching thermal equilibrium;

$T$—temperature, i.e., control parameter;

$I$—pre-set number of iterations to be performed.

The steps of the model are as follows. First, representatives of service zones, as potential locations of CDPs, are determined by applying an algorithm for determining one median in each zone [91]. After that, the "greedy" heuristic algorithm proposed in paper [86] is applied to the resulting network of potential locations. The algorithm is defined as follows [82].

At the beginning, the set of nodes in which the medians are located is empty. First, the problem of one median is solved and the obtained solution is included in the set of medians. The median can be determined by applying the already-mentioned algorithm from paper [91]. In each subsequent step, one new node, the inclusion of which reduces the value of the total distance the most, is included in the set of medians.

The solution of the greedy heuristic is loaded as the initial solution in the simulated annealing algorithm $m$ and added to the set $M$. Then, the same solution is loaded as the best solution. Then, in each iteration, an arbitrary solution $m'$ ($m' \notin M$) is chosen in the neighborhood of the current solution m. Depending on the objective function, the solution $m'$ becomes the best and/or current.

If the objective function of the solution $m'$ is not less than the objective function of the solution $m$, the value of the function $P$ is calculated, according to the Formula (7), and the number $R$ is generated and compared with $P$. If $P > R$ or the objective function of solution $m'$ is smaller than the objective function of solution $m$, solution $m'$ becomes the current solution ($m = m'$). If $P \leq R$, solution $m'$ is rejected, and $m$ remains the current solution.

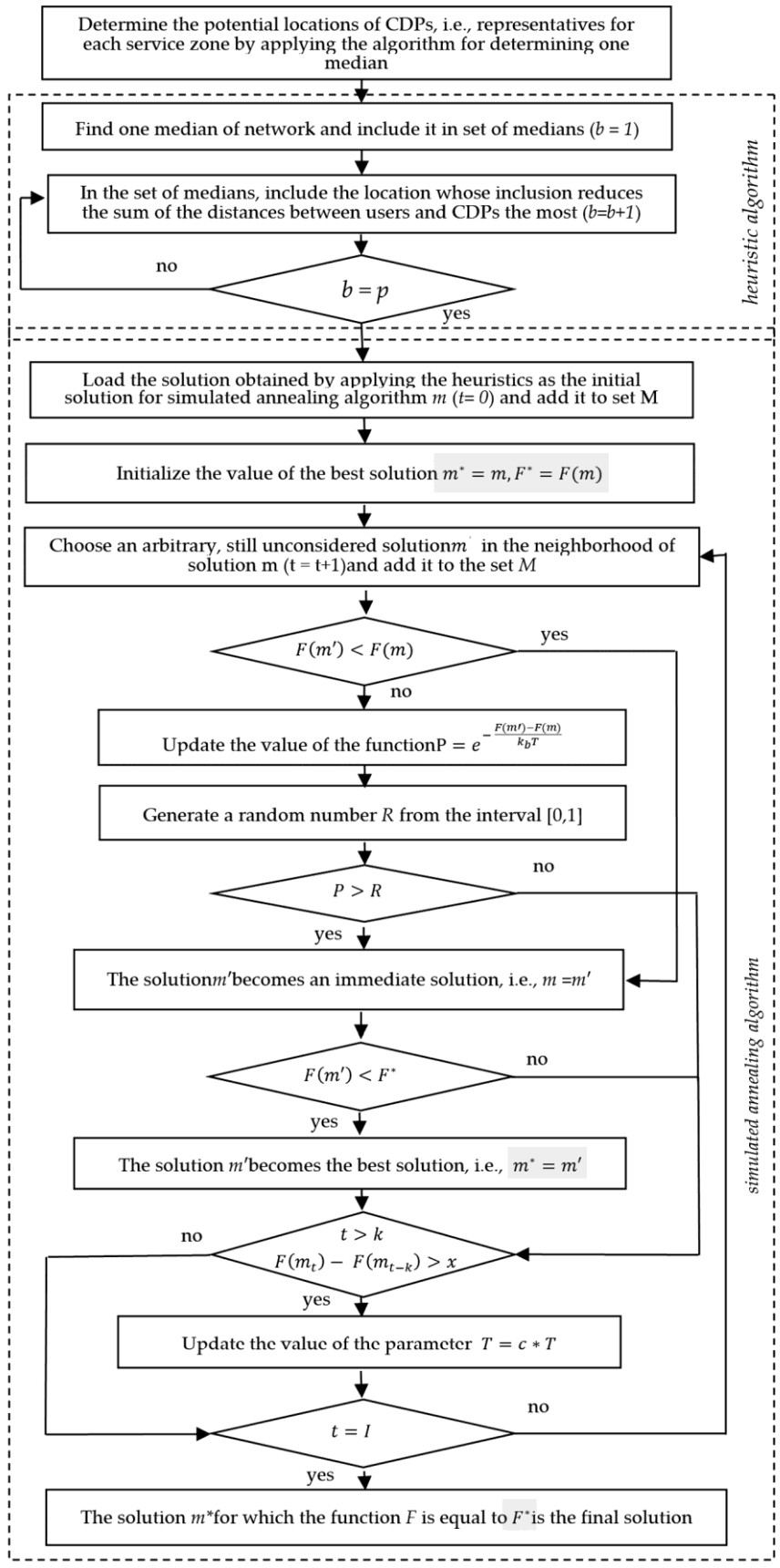

**Figure 1.** Proposed model.

After that, if the objective function of solution $m'$ is less than the objective function of the best solution at that time, solution $m'$ becomes the best solution ($m^* = m'$). Otherwise, the best solution remains unchanged.

The simulated annealing algorithm is implemented through a predefined number of iterations $i$, but is run only once. It is defined in such a way that at every $k$-th iteration of the algorithm, it is checked whether there has been a significant decrease in the criterion function (greater than $x$). If so, the algorithm searches (generates) solutions at the same temperature. If not, the temperature is automatically reduced and the search for solutions continues at the new temperature. Therefore, in both cases there is no user intervention, but the algorithm works by itself. In this way, it is ensured that the current best solution is always with an equal or smaller criterion function compared to the previous one.

## 4. Locating CDPs in Users' Households in the City of Belgrade

The proposed model was tested below on the example of a company from Belgrade. The company delivers non-food, non-perishable goods to the home addresses of users located in 956 zones. Figure 2 shows the representatives of each of the zones, obtained by applying the algorithm for determining one median, which was defined in paper [91].

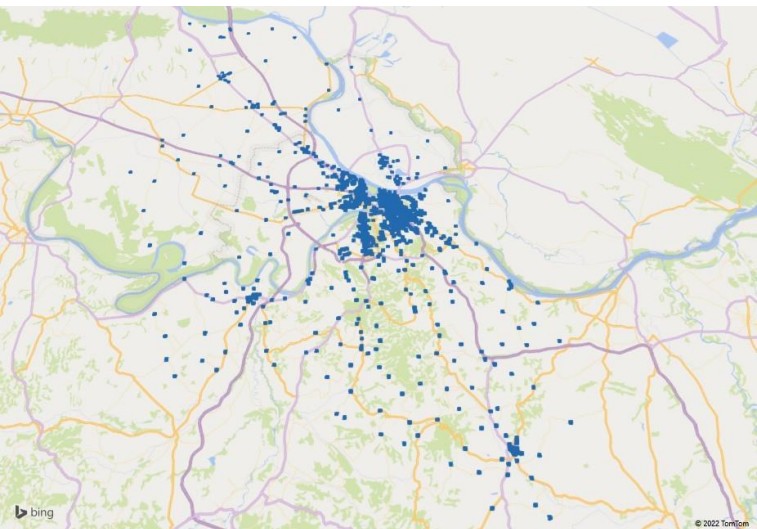

**Figure 2.** Representatives of service zones.

The company plans to introduce a new delivery model, delivery to CDPs, with the aim of reducing flows, the number of engaged workers and vehicles, the number of vehicle starts, energy consumption and costs, as well as attracting users with a service that is cheaper than the home delivery service. The company wants to use the households of its users as CDPs, with an appropriate monetary or other type of compensation (e.g., reduced price of goods, free delivery, etc.), in order to avoid the costs of investments in CDPs. It is necessary to determine the locations of households that should play the role of CDPs.

In order to solve the defined problem, data on the number of deliveries of a real-life company to each zone for 20 days were used. The total number of deliveries during this period was 31,857, so the average daily number of deliveries is equal to 1577.

In this example, it is assumed that the operator has complete freedom in making a decision on the number of CDPs, i.e., that there are no restrictions on the number of available households and their storage capacity. In addition, there are no user preferences in terms of the distance of CDPs. Let the number of CDPs whose location needs to be determined be equal to 50. This means that on average, 30 shipments/day will be stored in one CDP.

Euclidean distances between households are used as input values in the model. They were obtained on the basis of geographical coordinates according to the formula:

$$d_{ij} = \sqrt{\left(x_i - x_j\right)^2 + \left(y_i - y_j\right)^2} \tag{8}$$

where $x_i$ represents the latitude of location of user $i$, and $y_i$ is the longitude of that location.

The model proposed in the previous section was transformed into the program code and implemented in MATLAB software (© 1994–2023 The MathWorks, Inc., Natick, MA, United States). The obtained solutions are shown in Figures 3 and 4. Figure 3 shows the zones in which the CDPs were located, and Figure 4 shows zones served by the CDPs (one CDP serves the zones whose representatives are marked with the same color) in the case of applying the heuristic algorithm. Given that the input values in the model are the coordinates of the user's location, expressed in degrees of separation of latitude and longitude, the obtained solution is expressed in the same units. The sum of the distances between users and CDPs (multiplied by the number of requests) is 525.3, which corresponds to a distance of 58,381.3 km.

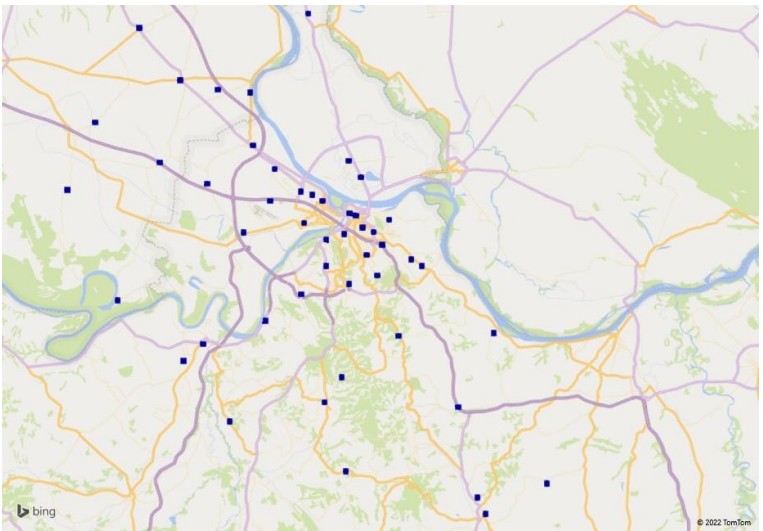

**Figure 3.** Locations of CDPs obtained using a heuristic algorithm.

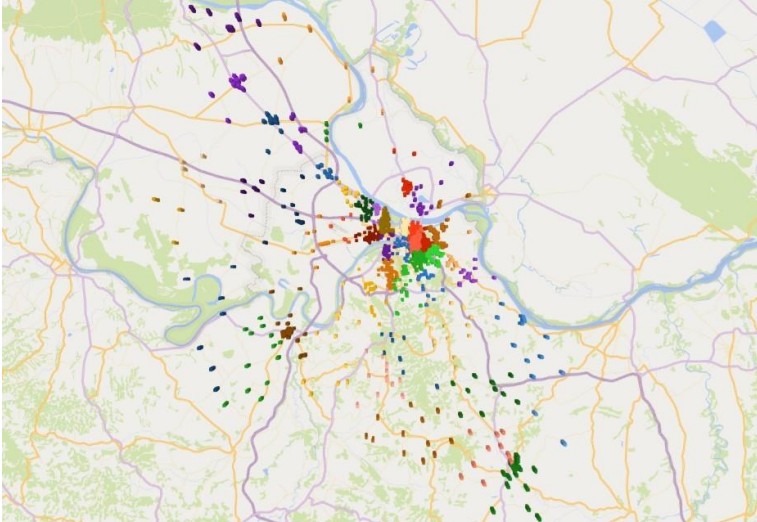

**Figure 4.** Allocation of users to CDPs using a heuristic algorithm.

The solution obtained by applying the heuristic algorithm represents the initial solution in the simulated annealing algorithm. The initial value of the parameter $T$ is set to 100. When thermal equilibrium is reached, the current temperature value is lowered by multiplying by the constant $c = 0.9$. The number of iterations is pre-defined and amounts to $I = 100,000$. The parameters $k$ and $x$ are assigned the values $k = 4$ and $x = 5$. The initial values of the simulated annealing parameters were defined based on examples from the literature [92], and then, through monitoring the operation of the algorithm, they were modified in order to improve its efficiency, i.e., the quality of the solution. Thus, the parameter $T$ is adjusted so that it is neither too high (then it takes more reduction to converge) nor too low (then points of potential global optimum could be exceeded) [93]. By applying the simulated annealing algorithm, the solution shown in Figures 5 and 6 was obtained. The sum of the distances expressed in degrees of separation is 496.7, which corresponds to a distance of 55,209.1 km.

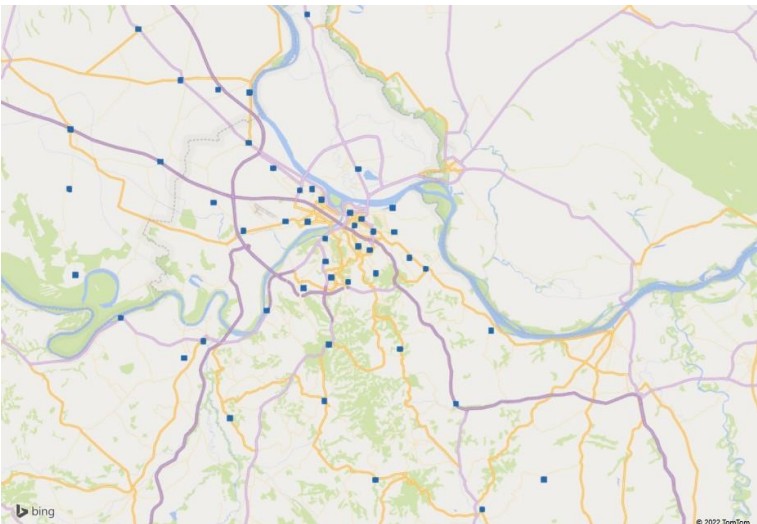

**Figure 5.** Locations of CDPs obtained using the simulated annealing algorithm.

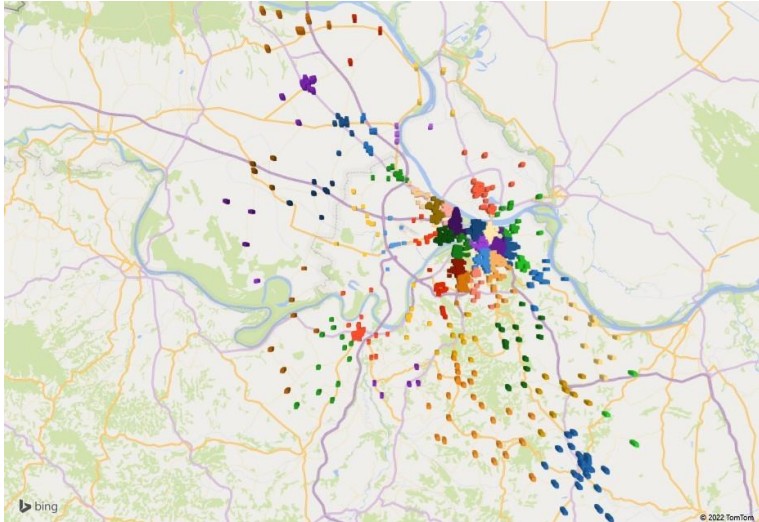

**Figure 6.** Allocating users to CDPs using the simulated annealing algorithm.

The defined model was developed and tested on PC Intel(R) Celeron(R) N4000 CPU @ 1.10 GHz with 4 GB RAM. The observed problem was solved in 15 min and 43 s of computer time, where the time of realization of the heuristic algorithm (about 7 s) is significantly

shorter than the time of realization of the simulated annealing algorithm, which is expected, considering the number of performed iterations of the second algorithm.

## 5. Discussion

The concentration of CDPs is slightly higher in the central city zones, both in the solution obtained by applying the heuristic algorithm and in the final solution. This is to be expected, bearing in mind that the largest number of users and thus potential locations for CDPs are located in these zones.

The working time of the computer is significantly longer in the case of applying the simulated annealing algorithm. Through the iterations of the application of this algorithm, periodic, very small reductions of the criterion function are present. In the final solution, 19 out of a total of 50 locations of CDPs selected during the application of heuristics were retained. The total distance was reduced by 3172.1 km, or about 5.4%.

In the final solution, the farthest user is 15.6 km away from the CDP in which he/she collects the goods. The minimum distance of the user from the CDP in which he/she collects the goods, excluding users whose households are CDPs (for which the distance is equal to 0 m), is 43 m. The average distance of users from the CDPs where they are served is about 2 km.

The decision on defining the number of locations that should be selected for CDPs is not argued in detail in the paper. The problem is set as non-capacitive, so this decision is not conditioned by storage capacity limitations, and in that context, it is subject to change. In this case, user allocation is carried out solely according to their distance from CDPs, which is also not limited by user preferences. However, user preferences can be subsequently taken into account and encourage the correction of the selected solution, by changing the number of CDPs that are selected.

Figure 7 shows the sum of the distances, and Figure 8 shows the average individual distances from users to the CDPs closest to them, in both cases depending on the different number of selected CDPs. Based on Figure 7, it can be concluded that the total distance decreases on average by about 1.48% with an increase in the number of CDPs by 1, i.e., by about 14.1% on average with an increase in the number of CDPs by 10.

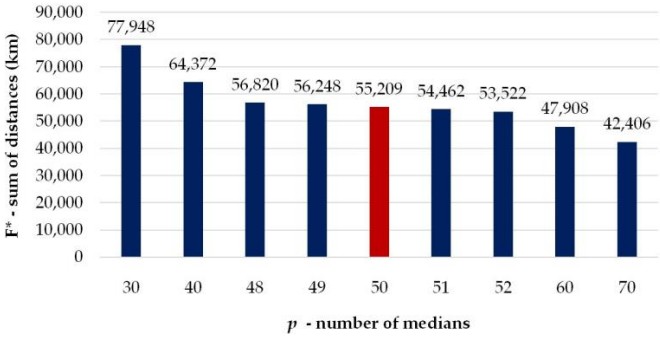

**Figure 7.** Changing the sum of the distances when modifying the number of medians $p$.

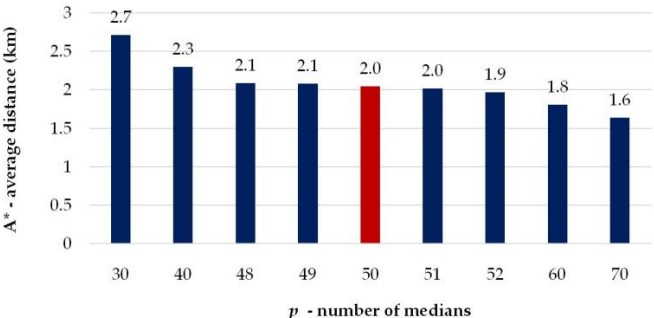

**Figure 8.** Changing average distance when modifying the number of medians $p$.

Compared to some European cities where the average distance of users from CDPs is equal to 1.6 km [10], in the obtained solution for $p = 50$, the distance is greater by 400 m. If the level of service quality in terms of the proximity of users to CDPs should be equalized with the practice of these European cities, the number of CDPs should be increased ($p > 50$) and around 70 CDPs should be located (Figure 8). The impact of the correction of the number of CDPs on delivery parameters (distance covered by suppliers and customers, time required to pick up goods, customer satisfaction, costs, etc.) should be analyzed in more detail in future research.

Correction of the number of CDPs according to the users' preferences is possible in the planning phase of the CDPs network, i.e., before the practical implementation of this concept. The current solution could be presented to the users as well as the opportunity to suggest changes in the distances. Moreover, the correction can be made after the implementation of the obtained solution, based on the users' statements about possible disadvantages observed when using CDPs services. In this case, we can talk about the so-called trial CDP [48], which, apart from this context, can also be significant for the collection of other, historical data important for the operator (information on the term of receipt of goods, possible simultaneous performing flows of goods taken from CDPs and other daily flows of users, used transport technology, etc.).

In paper [94], the authors indicated a significant discrepancy between theory and practice, i.e., the desired and existing level of service of CDPs on the example of the city of Rome. Namely, in order to achieve the desired distance between users and CDPs of 0.5 km, the city should be covered with a total of 5140 CDPs, and their current number is 192 [95]. From this it follows that the construction of additional CDPs would require high costs and significant space. In these conditions, the justification of applying the concept described in this paper is even more pronounced and obvious.

In a theoretical sense, the proposed model indicates the possibility of combining heuristic and metaheuristic algorithms in order to locate logistics nodes in the users' facilities. The goal was not to obtain results that compete with similar models in the literature, but to establish a framework and obtain satisfactory results for the defined problem. In this sense, locating CDPs based on household use can be realized in future research using other algorithms, in order to compare the results and efficiency of different approaches. In addition, it provides a basis for upgrading, modifying, or developing similar theoretical models for solving the same or similar problems. Finally, the defined model can be applied in a proposed or modified form in the logistics practice of companies for designing or redesigning distribution networks (CDPs or other facilities on the network). The implementation of solutions obtained by applying such models can bring significant economic benefits to companies.

## 6. Conclusions

A new concept of using the households as CDPs for making collections and deliveries of goods for users in an urban area is introduced in this paper. The main problems in this concept are the selection of appropriate CDP locations and the allocation of users to them. In order to solve this problem, a novel hybrid model for determining $p$-median is defined, based on the heuristic "greedy" algorithm and the simulated annealing algorithm. The goal was to locate CDPs in users' households, allocate other users to each of the selected locations, and minimize the sum of the distances between users and the nearest CDPs. The model was tested on the case of locating CDPs in the city of Belgrade. In addition, the modifications were made to the number of CDPs, which affects the differences in the level of service from the aspect of the distance of the user from the CDPs. In this way, the main goal of the paper was achieved, and we established a basis for further theoretical development and practical application of the model in the described or similar problems.

The given problem and the proposed model for solving it are defined in the context of many new or relatively new phenomena (e-commerce, CDPs, crowd logistics), which are still characterized by numerous challenges and questions that need to be answered.

Additionally, the results of applying the proposed model do not provide information about the effects of the introduction of CDPs, which would make its theoretical and practical implications much clearer and more complete. That is why it is necessary to conduct additional research on various aspects of this problem.

It is necessary to analyze the information aspect of the application of crowd storage in the functioning of CDPs, as well as the potential and adequacy of households to perform the role of CDPs. Another direction of future research is the analysis of the effects of the defined model application, i.e., the comparative analysis of delivery parameters (traveled distance, routing efficiency, costs, etc.) before and after the introduction of CDPs. The focus of attention should be on the comprehensive recording and research of the daily flows of users and their tendency to collect goods from CDPs during the implementation of these flows. Attention should also be paid to comparative analysis of the logistical, economic, and other parameters generated by additional flows from the collection of the goods. It is necessary to carry out a comprehensive analysis of the factors influencing the number, location, and efficiency of CDPs and the allocation of users. The defined model can be modified in terms of limiting the number and spatial dispersion of households available for CDPs, their storage capacity, priorities in decision making, the way of determining the distance from users to CDPs (time or real distance instead of Euclidean), etc. A comparative analysis of the results of the application of the proposed and modified models could also be the focus of future research. Moreover, future studies should investigate other algorithms, in order to compare the results and efficiency of different approaches against this one. Finally, it is necessary to analyze the differences in input (number, territorial dispersion and density of users, storage capacity of households) and output parameters (distance of users from CDPs, other logistical parameters, economic parameters) when locating CDPs in urban and rural areas.

**Author Contributions:** Conceptualization, Ž.S., S.T., M.K. and M.V.; Formal analysis, Ž.S., S.T., M.K. and M.V.; Methodology, S.T., M.K. and M.V.; Writing–original draft, Ž.S., S.T., M.K. and M.V. All authors have read and agreed to the published version of the manuscript.

**Funding:** This research received no external funding.

**Data Availability Statement:** The data used are owned by the company. They may be made available with the company's permission.

**Conflicts of Interest:** The authors declare no conflict of interest.

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
