# Peer review of "Locating Collection and Delivery Points Using the p-Median Location Problem"

_logistics, 2022_

Round 1

Reviewer 1 Report

The study is on solving collecting and delivery points for home delivery services formulating as a p-median location problem. Authors use hybird form of heuristic and metaheuristic approaches. There are some points should be cleared:

1. There is no evidence that linear programming model fails to obtain optimal results and we need heuristics.

2. It is not clear how the parameters in simulated annealing are obtained or optimized. Are they result of a Desing of Experiment study?

3. We cannot see any results about the efficiency of the selected solution approach.

Author Response

Comment #1: There is no evidence that linear programming model fails to obtain optimal results and we need heuristics.

Response: The paper does not state that the optimal solution cannot be obtained by applying the linear programming model, but that it is more difficult to reach it that way, primarily in terms of the operating time. This is now highlighted in the paper in the fifth paragraph of Section 3 (Methodology).

Comment #2: It is not clear how the parameters in simulated annealing are obtained or optimized. Are they result of a Desing of Experiment study?

Response: We have additionally explained this in the penultimate paragraph of the Section 4.

Comment #3: We cannot see any results about the efficiency of the selected solution approach.

Response: A part related to this has been added to the paper (last paragraph of the Section 4) according to your suggestions.

Reviewer 2 Report

The authors should have compared their algorithm with another one from the literature. Being a very studied problem, the possibility of finding an algorithm that improves those existing algorithms is small. Moreover, the improvement obtained is also expected to be small.

The analysis of the impact of the number of CDPs on the sum and average distances from the customers is of great value. More than the novelty of the results, it is the quantification of this impact.

On p.7, describing the algorithm, if the solution m’ is selected randomly in the neighborhood of m, it is not clear why the algorithm would not cycle, selecting again the same solution. It is expected that there is a memory of visited solutions, but I think it is not stated. Furthermore, the neighborhood is not described. Maybe it is selected from the set of customers in that area.

Also, in Figure 1, in the first activity of simulated annealing, it is referred to as the simulated hardening algorithm. This variant is slightly different from the standard one but is the only place in the document where it is mentioned. It should be described or explained why it is used. In the same figure, I do not know why the condition for comparing ?(??) − ?(???) = ?  should be >x instead.

In p.8, why use the probability of choosing worse solutions in the work by Teodorović (2016) and not the one by Golden and Skiscim (1986). Also, it should be explained why the Boltzmann parameter is chosen to one.

When showing the results, the number of runs of the algorithm to obtain those results is not clear. As probabilities of selecting worse solutions when there is no progress in the objective function, the results should be different in different runs of the algorithm.

In Section 5. The results are compared only with results of other authors in other cities, but not with other algorithms or with other authors in the literature. The effort in developing this algorithm for a well-known problem should be supported with some comparisons.

Minor corrections are as follows.

In p.6, the notation is clearly defined, so when c is defined, T should also be defined as Temperature.

In Section 4. Figure 2 is referred to as Figure 1 in the text.

In Figure 4, the map is slightly different.

Author Response

Comment #1: The authors should have compared their algorithm with another one from the literature. Being a very studied problem, the possibility of finding an algorithm that improves those existing algorithms is small. Moreover, the improvement obtained is also expected to be small.

Response: The main focus of this paper was on the establishment of a new concept in the area of city logistics, which is the use of households as CDPs. In addition we have developed a new model to solve the problem of locating CDPs and allocating the users to them. Therefore, the goal of the model was not competitive efficiency compared to other models, but solving the defined problem. This is additionally explained in the last paragraph of Section 5.

Having this in mind, we have proposed solving the same problem using other algorithms, in order to compare the results and efficiency of different approaches in some future studies (last paragraph of Section 6).

Comment #2: The analysis of the impact of the number of CDPs on the sum and average distances from the customers is of great value. More than the novelty of the results, it is the quantification of this impact.

Response: Thank you for this remark. We have additionally highlighted this in the Abstract (Results), Introduction (penultimate paragraph) and Conclusion (first paragraph).

Comment #3: On p.7, describing the algorithm, if the solution m’ is selected randomly in the neighborhood of m, it is not clear why the algorithm would not cycle, selecting again the same solution. It is expected that there is a memory of visited solutions, but I think it is not stated. Furthermore, the neighborhood is not described. Maybe it is selected from the set of customers in that area.

Response: The model remembers previously generated solutions, which is now pointed out in the paper (in the description of notation - 13th paragraph of Section 3, in the Figure 1 and in the 16th paragraph of Section 3).

Also, a clarification of the term solution in the neighborhood is given (in the description of notation - 13th paragraph of Section 3 in the description of notation - 13th paragraph of Section 3).

Comment #4: Also, in Figure 1, in the first activity of simulated annealing, it is referred to as the simulated hardening algorithm. This variant is slightly different from the standard one but is the only place in the document where it is mentioned. It should be described or explained why it is used. In the same figure, I do not know why the condition for comparing (?t) − (??−?) = ?  should be >x instead.

Response: The term “simulated hardening” is used by mistake. We meant simulated annealing. We have corrected this.

You are right. We have corrected this and it is now (?t) − (??−?) > ?.

Comment #5: In p.8, why use the probability of choosing worse solutions in the work by Teodorović (2016) and not the one by Golden and Skiscim (1986). Also, it should be explained why the Boltzmann parameter is chosen to one.

Response: We have explained this in the 9th and 10th paragraphs of Section 3 (equations (6) and (7) and accompanying text). First the formula from the paper (Golden & Skiscim, 1986) is presented, and then the derived formula is presented, in which Boltzmann parameter is equal to 1 and which is used in many studies (Schneider & Kirkpatrick, 2007; Filippone et al., 2011, Teodorović, 2016) in order to simplify the mathematical expression and facilitate the implementation of the algorithm.

Comment #6: When showing the results, the number of runs of the algorithm to obtain those results is not clear. As probabilities of selecting worse solutions when there is no progress in the objective function, the results should be different in different runs of the algorithm.

Response: This is additionally explained in the last paragraph of the Section 3.

Comment 7: In Section 5. The results are compared only with results of other authors in other cities, but not with other algorithms or with other authors in the literature. The effort in developing this algorithm for a well-known problem should be supported with some comparisons.

Responses: The main focus of this paper was on the establishment of a new concept in the area of city logistics, which is the use of households as CDPs. In addition we have developed a new model to solve the problem of locating CDPs and allocating the users to them. Therefore, the goal of the model was not competitive efficiency compared to other models, but solving the defined problem. This is additionaly explained in the last paragraph of Section 5.

Having this in mind, we have proposed solving the same problem using other algorithms, in order to compare the results and efficiency of different approaches in some future studies (last paragraph of Section 6).

Comment #8: Minor corrections are as follows.

  • In p.6, the notation is clearly defined, so when c is defined, T should also be defined as Temperature.
  • In Section 4. Figure 2 is referred to as Figure 1 in the text.
  • In Figure 4, the map is slightly different.

Response: We have made all necessary corrections.

Reviewer 3 Report

This paper proposed a hybrid model for locating CDPs in users’ households, and the model was tested on the case of locating CDPs in the city of Belgrade. The model is new and interesting. There are some problems in this manuscript.

1.     p-median is a classical logistic model. Are there any better models to solve the problem?

2.     There are many machine learning methods to solve the multi-objective problem. What is the algorithm to solve the p-median model? The algorithm has to be described.

3.     There lacks sufficient literature. Many newest research works have to be added.

4.      The contribution of this work has to be given out clearly.

Author Response

Comment #1: p-median is a classical logistic model. Are there any better models to solve the problem?

Response: The main focus of this paper was on the establishment of a new concept in the area of city logistics, which is the use of households as CDPs. In addition we have developed a new model to solve the problem of locating CDPs and allocating the users to them. Therefore, the goal of the model was not competitive efficiency compared to other models, but solving the defined problem. This is additionaly explained in the last paragraph of Section 5.

Having this in mind, we have proposed solving the same problem using other algorithms, in order to compare the results and efficiency of different approaches in some future studies (last paragraph of Section 6).

Comment #2: There are many machine learning methods to solve the multi-objective problem. What is the algorithm to solve the p-median model? The algorithm has to be described.

Responses: We have additionally elaborated these issues in Figure 1 and in the accompanying text (from notation to the end of Section 3). The given step descriptions of the greedy heuristic and the simulated annealing algorithm are similar or slightly simplified compared to the flowcharts/pseudocodes given in the literature:

Kuehn, A. A., & Hamburger, M. J. (1963). A Heuristic Program for Locating Warehouses. Management Science, 9(4), 643–666.

Rere, L. R., Fanany, M. I., & Arymurthy, A. M. (2015). Simulated annealing algorithm for deep learning. Procedia Computer Science, 72, 137-144.

Al-khedhairi, A. (2008). Simulated annealing metaheuristic for solving p-median problem. Int. J. Contemp. Math. Sciences, 3(28), 1357-1365. 

Comment #3: There lacks sufficient literature. Many newest research works have to be added.

Responses: We have supplemented the literature review with more recent works.

Comment #4: The contribution of this work has to be given out clearly.

Responses: The contributions of the paper are now further clarified in the Abstract (Results), Introduction (penultimate paragraph) and Conclusion (first paragraph).

Round 2

Reviewer 2 Report

Thank you for correcting and clarifying all my suggestions. The article has been greatly improved. There is only a small mistake in the last paragraph of Section 3. There is a mistake or omission in the third line of this paragraph. Also, in some of the references, the years appear close to the title of the reference.

Author Response

Comment #1: Thank you for correcting and clarifying all my suggestions. The article has been greatly improved.

Response: We thank you for this affirmative comment.

Comment #2: There is only a small mistake in the last paragraph of Section 3. There is a mistake or omission in the third line of this paragraph.

Response: Thank you for noticing this. Somehow in the reviewing process important part of the paragraph got deleted. We have added the missing part of the paragraph.

Comment #3: Also, in some of the references, the years appear close to the title of the reference.

Response: We have corrected this. We also noticed missing spaces between some words, which we also corrected.